# Obesity Interventions for Aboriginal and Torres Strait Islander Children and Adolescents: A Scoping Review of Impact and Outcomes

**DOI:** 10.3390/ijerph22111671

**Published:** 2025-11-03

**Authors:** Kabita Kharka, Kristina Zafirovski, Fahad Hanna

**Affiliations:** Department of Health and Education, Torrens University, Melbourne, VIC 3000, Australia; kabita.kharka@health.torrens.edu.au (K.K.); kristina.zafirovski@torrens.edu.au (K.Z.)

**Keywords:** obesity prevention, indigenous, children, Australia, interventions, summary of evidence

## Abstract

**Background**: Childhood Obesity is a significant and growing Public Health threat among Aboriginal and Torres Strait Islander (ATSI) children and adolescents in Australia. Health sectors in Australia have been focusing on health intervention programs across various states to prevent childhood obesity. This review aims to analyse the impact of obesity intervention programs conducted among children and adolescents of ATSI communities across Australia and report on the best practices for conducting future research. **Objectives:** This scoping review synthesised existing literature on the obesity prevention interventions programs among ATSI Children and Adolescents in Australia and examined their scope, implementation, and outcomes; evaluated their cultural appropriateness; and highlighted critical enablers and barriers. **Methods**: This scoping review analysed scholarly journal articles that reported on the findings of obesity intervention programs delivered across ATSI children and adolescents. Only Quasi-experimental and Randomized Control Trials (RCTs) were selected for the review. A full search has been carried out in Health databases such as Cochrane Library, Medline, PubMed, and ProQuest Central for the past 15 years. The framework of the Joanna Briggs Institute (JBI) for Scoping reviews was followed throughout this review. **Results**: Eleven studies met inclusion criteria. Findings were clustered into five themes: (i) community-led, systems-based interventions improved health behaviours and anthropometry; (ii) culturally tailored, community-embedded programs enhanced engagement and health literacy; (iii) early childhood and family-focused approaches showed promise; (iv) community or policy initiatives yielded mixed results; and (v) behavioural interventions were ineffective without addressing social and structural determinants. The most effective programs were community-delivered, multi-stakeholder, and centred on empowerment and capacity building. **Conclusions**: Obesity prevention efforts targeting Aboriginal and Torres Strait Islander children and adolescents are most successful when community-led, culturally grounded, and supported by multiple stakeholders. These findings underscore the necessity of culturally sensitive, participatory approaches. Further research is needed to strengthen the evidence base and inform sustainable, policy-relevant strategies for childhood obesity.

## 1. Introduction

Childhood obesity among Aboriginal and Torres Strait Islander (ATSI) children and adolescents remains a critical and persistent public health concern in Australia. Indigenous children experience significantly higher rates of overweight and obesity compared to their non-Indigenous peers [1]. According to the Australian Institute of Health and Welfare, approximately 38% of ATSI children aged 2–17 years were living with overweight or obesity in 2018–2019, highlighting an ongoing and substantial health disparity [1].

This issue is deeply interconnected with broader social determinants of health (SDH), such as socioeconomic disadvantage, systemic barriers to healthcare access, colonisation, intergenerational trauma, and inadequate environmental infrastructure [2,3,4,5].

Obesity is a major contributor to the widening health gap between Indigenous and non-Indigenous Australians, accounting for an estimated 16% of the overall disparity and representing the second-largest contributor to this gap [6]. Social determinants collectively contribute to approximately 34% of this inequality, with factors such as dispossession of land and culture, imposed welfarism, systemic racism [7], and limited access to culturally appropriate healthcare services continuing to impact community wellbeing [2,6,8]. Despite growing national attention to childhood obesity in ATSI communities, many early intervention programs have primarily focused on knowledge dissemination without adequately addressing the economic, cultural, and social realities that shape health behaviours [9,10,11]. As a result, such interventions have often failed to produce meaningful and sustained outcomes [12].

Research shows that early-life risk factors, including socioeconomic disadvantage and maternal behaviours [13], significantly contribute to the development of obesity among ATSI children [14]. Dietary behaviours characterised by high consumption of sugar-sweetened beverages and foods rich in saturated fats, alongside low intake of fruits and vegetables, further exacerbate the risk [4]. While these risk factors are well recognised, regional disparities in research remain evident, with relatively few studies conducted in states such as Victoria compared to other regions of Australia [15]. This limits the generalisability and applicability of existing findings and interventions across different communities.

National strategies have increasingly acknowledged the need for culturally safe, community-led health promotion programs [16,17]. Community-based initiatives such as the *Romp & Chomp* early-childhood program demonstrated the potential of local, multi-sector partnerships to improve dietary behaviours and reduce obesity rates in young children [18]. The *Close the Gap* campaign highlights the importance of addressing systemic inequities and social determinants to improve Indigenous health outcomes and reduce the burden of chronic disease [19]. Likewise, the National ATSI Health Plan 2021–2031 identifies childhood obesity prevention as a national priority and calls for culturally responsive, community-driven approaches that address both individual behaviours and structural barriers to health [20]. Earlier national health promotion efforts such as the *Go for 2&5* campaign also aimed to improve dietary behaviours and fruit and vegetable intake at the population level [21]. Despite these commitments, current obesity prevention efforts require more comprehensive evaluation, particularly regarding their effectiveness, sustainability, and cultural responsiveness.

Previous research has laid important groundwork and influenced health policy development, yet gaps remain in understanding how interventions can be adapted to the unique contexts of Indigenous communities [22]. This scoping review seeks to explore the existing evidence on obesity prevention initiatives for ATSI children and adolescents and to identify the factors that contribute to the success or limitations of these programs. This review examines the scope, implementation, and outcomes of current programs, while also evaluating their cultural appropriateness and identifying key enablers and barriers. It provides evidence to guide future health promotion initiatives and policy, emphasising the need for interventions that are evidence-based, culturally safe, sustainable, and responsive to the lived realities of Aboriginal and Torres Strait Islander children and their families. Through this approach, it contributes to addressing existing knowledge gaps and may serve as a precursor for more targeted systematic reviews or evaluations, assisting policymakers and practitioners in planning effective, community-centred interventions [23].

## 2. Methodology

A comprehensive scoping review was conducted following the Joanna Briggs Institute (JBI) methodology for scoping reviews, which is a structured search framework initially proposed by Arksey and O’Malley in 2005 [24,25,26]. The scoping review involves several key steps: formulating the research question, locating relevant literature, selecting studies that meet the inclusion criteria, extracting the findings, and presenting the results [25]. The choice of a scoping review methodology aligns with the aims and objectives of this research, as it allows for a broad exploration of existing evidence while ensuring methodological rigor, transparency, and reproducibility. The review process adhered closely to the JBI guidelines and the Preferred Reporting Items for Systematic Reviews and Meta-Analyses- extension for Scoping Reviews (PRISMA-ScR) recommendations for reporting [27,28]. An iterative search strategy was implemented, guided by the following central research question: “What is the impact of health intervention programs on reducing childhood obesity among ATSI children and adolescents in Australia?”

### 2.1. Database Search

Literature search was conducted on major databases such as ProQuest Central, Cochrane, PubMed Central and Medline to identify papers published between January 2010 and January 2025. The Boolean search was used with specific keywords and synonyms like ‘aboriginal children’ AND ‘Indigenous child’, AND (health intervention programs or HEALTH PROMOTION) AND (obesity) OR, ‘intervention OR ‘knowledge’,’ obesity AND ‘Aboriginal children’, AND ‘Health intervention. AND (Australia*) AND (CLINICAL TRIALS). Multiple databases were chosen for this study to improve results and reduce the risk of overlooking any eligible studies that could be used during our final appraisal [29].

### 2.2. Choice of Design

A scoping review was chosen because it allows for a broad and comprehensive search strategy while ensuring reproducibility, transparency, and reliability of the existing evidence in the field [25,30]. This approach is particularly valuable for examining the impact of health interventions on obesity among Indigenous children. Additionally, scoping reviews help to map key concepts within a research area, clarify working definitions, and establish the boundaries of the topic [31]. Moreover, a scoping review of scoping reviews found that their primary purposes include understanding the extent of available literature, summarizing existing evidence, and guiding future research directions [25]. This research team, as part of the broader research unit, has adopted and progressively refined the scoping review methodology in recent years, applying it successfully to summarise evidence on important public health issues. This established experience strengthens the rationale for using the scoping review design in the present study [32,33,34].

### 2.3. Data Sources

A systematic search was performed on major public health databases, including the Cochrane Library, MEDLINE, PubMed, and ProQuest Central. Additionally, relevant online journals were searched to capture the most recent publications. The selection criteria included peer-reviewed journal articles, health intervention studies, quantitative research designs, and a focus on Australian ATSI schools [35]. This review specifically focused on two common quantitative research designs: randomized controlled trials and quasi-experimental studies [36].

### 2.4. Study Selection

After titles and abstracts of studies were screened based on the inclusion criteria, eligible articles were identified for full-text review. Subsequently, the full texts of these selected articles were examined to confirm their inclusion. Additionally, the reference lists of all included articles were searched to identify any further relevant studies. Only original studies published within the last 15 years were included. Articles not written in English, editorials, reviews, and unpublished studies were excluded. Bibliographic references were compiled for all relevant articles [37]. We included peer-reviewed studies that provided original data such as randomised controlled trials and quasi-experimental studies. After data extraction was completed, Braun and Clarke’s approach to thematic analysis was used to evaluate the data. The approach consisted of six steps: 1. Being familiar with the data; 2. Producing initial codes for the data; 3. Searching for potential themes; 4. Reviewing themes; 5. Defining and naming themes; and 6. Reporting and analysing themes [31]. Phase 6 was completed using PRISMA-ScR guidelines [28,38]. Compliance with the PRISMA-ScR reporting guidelines is detailed in Appendix A.

### 2.5. Data Charting

Following the 2022 JBI protocol, the data extraction and charting process provide essential insights for readers, offering a comprehensive overview of the outcomes relevant to the scoping review questions [26]. After completing the screening phase, the data extraction process was initiated. A data charting table was used, and data were independently extracted by two reviewers (K.K. and K.Z.). Each study was assessed against the predefined eligibility criteria for inclusion or exclusion in the review, with decisions based on consensus between the two primary reviewers. In cases where agreement could not be reached, a third reviewer (F.H.) was consulted to resolve discrepancies and make the final determination.

Looking at the data extraction table (Table 1), the main information included records for each source. This included the reference and any results or findings that were relevant to the research question [26]. During the review stage, the charting table was revised accordingly. The selected studies for our scoping review were then organised based on the author, year of publication, age group/sample size, state, duration of intervention and details of interventions, and key focus area. Both reviewers reached a complete consensus on all the extracted data. The charting table was consistently revised to facilitate ongoing data enhancement.

### 2.6. Thematic Analysis

Secondary data analysis offers several distinct advantages, particularly in public health research. One major benefit is its cost-efficiency, as the data have already been collected and are typically available in digital formats. This allows researchers to devote more time and resources to interpreting the data rather than undertaking time-consuming and often expensive data collection procedures [49]. Additionally, secondary data sets often provide access to large, representative samples that may not be feasible to collect independently, supporting broader and more generalisable insights.

Thematic analysis, a widely used and valuable qualitative method, was employed in this review to systematically identify and categorise patterns or themes within the selected studies [31]. The data analysis process involved identifying significant themes emerging across the reviewed literature, guided by the principles of thematic analysis [50]. These findings were organised into four major themes reflecting commonalities in obesity intervention approaches.

As all included studies were either RCTs, cluster RCTs, or quasi-experimental designs, some limitations inherent to cluster sampling—such as reduced diversity within sample populations—may have influenced the generalisability of results [36]. The scoping review process was also conducted with reference to the Critical Appraisal Skills Programme (CASP) checklist, ensuring methodological rigour and transparency. A total of five themes were identified for this analysis.

### 2.7. Quality Appraisal of Included Studies Using CASP Checklist

To assess the methodological quality of the included studies, we applied the Critical Appraisal Skills Programme (CASP) checklist for randomized controlled trials and quasi-experimental designs. The appraisal revealed that while most studies addressed clearly focused issues and measured outcomes reliably, limitations were noted in randomization, blinding, and follow-up completeness. Overall, the risk of bias was rated as moderate for the majority of studies, with one study assessed as high risk. These findings underscore the need for more rigorously designed interventions in future research. A summary of the CASP assessment is presented in Table 1.

## 3. Results

A total of 757 records were identified through five databases: ProQuest (*n* = 263), Cochrane (*n* = 24), PubMed (*n* = 6), Medline (*n* = 455), relevant Aboriginal health journals (*n* = 7) and citation searching (*n* = 2). Following the removal of 227 duplicates, 530 articles remained for further screening based on titles and abstracts. A preliminary eligibility screening (language, publication date, document type, and relevance based on title/keywords) excluded 452 irrelevant records. The remaining 78 records underwent formal title and abstract screening against predefined inclusion criteria, focusing on obesity intervention or prevention programs among ATSI children in Australia. Of the 78 records screened, 55 were excluded because the focus was not on obesity prevention. The 23 remaining full-text articles were assessed for eligibility. Twelve of these were excluded for not being randomized controlled trials (RCTs) or quasi-experimental studies. In total, 11 studies were included in the final review. The full screening process is illustrated in the PRISMA flow diagram (Figure 1).

A total of eleven studies [9,39,40,41,42,43,44,45,46,47,48] met the inclusion criteria for this scoping review, comprising both randomized controlled trials and quasi-experimental designs conducted across various Australian states. These studies targeted Aboriginal and Torres Strait Islander children and adolescents, with interventions ranging from school-based programs to community-led initiatives. Table 2 provides a detailed overview of each study, including author, year, location, sample characteristics, intervention duration and design, key parameters assessed, and summary of findings. This table highlights the diversity of approaches and outcomes across the included studies, offering valuable insights into the scope and implementation of obesity prevention efforts in Indigenous communities. Table 3 provides a visual summary linking each included study to its reported outcomes and the thematic category identified in the scoping review. Where available, brief statistical results are included to quantify impacts.

### 3.1. Study Themes

#### 3.1.1. Community-Led, Systems-Based Interventions Improve Health Behaviors and Anthropometry

Two studies emphasized the value of community-designed, systems-based interventions in promoting healthier behaviours and improving BMI outcomes among Indigenous children [39,48]. The RESPOND trial demonstrated significant improvements in Body Mass Index (BMI), Health-Related Quality of Life (HRQoL), and behaviours such as screen time, water intake, and physical activity, emphasizing the power of embedding local ownership into prevention systems [39]. The Eat Well Be Active program showed moderate BMI reductions, especially among younger children, further reinforcing the importance of early, sustained, and multi-strategy interventions [48].

#### 3.1.2. Culturally Tailored and Community-Embedded Programs Enhance Engagement and Health Literacy

Three programs designed with cultural relevance and community involvement, especially for Indigenous and minority groups, were highly effective in promoting engagement, knowledge, and behavioural shifts [44,45,46]. The Good Start Program, tailored to Māori and Pacific Islander communities, increased nutrition knowledge and physical activity through culturally embedded performing arts [45]. Similarly, the Deadly Choices program improved health behaviours and literacy among ATSI adolescents in a short 7-week period [44]. The Sporting Chance Program also demonstrated strong physical activity improvements and high acceptability by leveraging Indigenous knowledge and community-school partnerships [46]. These studies underline the critical role of cultural relevance and empowerment in engaging ATSI youth and driving meaningful behaviour change.

#### 3.1.3. Early Childhood and Family-Focused Interventions Yield Promising Results

Two interventions targeting young ATSI children and their families showed promising results though modest impacts on nutrition and anthropometry [40,47]. The Baby Teeth Talk study offered early dietary advice alongside oral health messaging, showing small positive dietary changes but no BMI impact [40]. The Fruit & Veg Subsidy Trial improved nutritional markers like haemoglobin and reduced antibiotic use, though it did not shift Body Mass Index (BMI) [47]. These findings suggest that family-based, early-life programs can improve foundational health behaviours and biological indicators and may require longer timelines or additional components to affect weight outcomes.

#### 3.1.4. Mixed Results for Impact of Community or Policy Intentions

Four programs showed limited or no significant impact on obesity-related outcomes, particularly BMI, despite being community-based or culturally relevant [40,41,42,43,47]. For example, the OPAL program [43] used a systems-wide model but lacked sufficient community ownership, leading to no significant changes in BMI or health behaviours. The Jump Start Trial [42], which targeted physical activity in preschoolers across disadvantaged communities, also failed to demonstrate positive outcomes, largely due to low implementation fidelity and variability in delivery. Similarly, the Baby Teeth Talk study [40], while focused on oral health and early dietary behaviours among ATSI children, did not result in measurable changes in BMI, though it did yield some minor dietary improvements. The Fruit and Vegetable Subsidy Trial [47] improved nutritional markers such as haemoglobin levels and reduced antibiotic use, but it did not impact overweight or obesity prevalence despite addressing food access.

#### 3.1.5. Behavioural Interventions—Lack of Social and Structural Determinants

This review found that while some programs improve individual health behaviors, only a limited number explicitly consider or address broader structural and environmental factors such as food insecurity, housing, access to healthcare, and other systemic inequities that influence obesity outcomes in ATSI children and adolescents. For example, Browne and colleagues [15] highlight how systemic inequities continue to impact disparities in weight and well-being, and the Fruit & Veg Subsidy Trial [47] demonstrates the challenges of improving BMI without addressing complex food environments. This suggests that behavioural interventions alone may be insufficient without complementary strategies targeting social and structural determinants.

A summary of study outcomes and their associated themes is detailed in Table 3.

**Table 3 ijerph-22-01671-t003:** Matrix of Study Outcomes and Thematic Categories.

Study	Outcomes	Theme
[39] Allender et al. (2024)	BMI reduction, HRQoL, screen time, water intake, physical activity	Community-led, systems-based
[40] Smithers et al. (2017)	Minor dietary improvements, no BMI impact	Early childhood, family-focused
[41] Waters et al. (2018)	Mixed results, limited BMI impact	Community or policy initiatives
[42] Okely et al. (2020)	No significant outcomes	Community or policy initiatives
[43] Bell et al. (2017)	No significant BMI or behaviour changes	Community or policy initiatives
[44] Malseed et al. (2014)	Improved health behaviours and literacy	Culturally tailored, community-embedded
[45] Mihrshahi et al. (2017)	Increased nutrition knowledge and physical activity	Culturally tailored, community-embedded
[46] Peralta et al. (2014)	Improved physical activity, high acceptability	Culturally tailored, community-embedded
[47] Black et al. (2013)	Improved haemoglobin, reduced antibiotic use, no BMI impact	Early childhood, family-focused
[48] Pettman et al. (2014)	Moderate BMI reductions	Community-led, systems-based
[15] Browne et al. (2022)	Structural barriers highlighted, no BMI impact	Behavioural interventions lacking structural determinants

## 4. Discussion

This scoping review examined the impact of obesity intervention programs aimed at preventing and reducing childhood obesity among ATSI children and adolescents in Australia. The findings suggest that community-led, systems-based interventions, particularly those emphasizing cultural tailoring, Indigenous leadership, and integration within broader health systems, are the most effective in improving health behaviours and achieving modest yet meaningful changes in Body Mass Index (BMI) outcomes [51].

Programs such as Good Start, Deadly Choices, and Sporting Chance demonstrate the importance of embedding cultural knowledge and community engagement into health promotion. These initiatives align with Indigenous values and ways of knowing, enhancing program acceptability, participation, and health literacy. Interventions targeting early childhood and families have also shown promise, particularly in promoting healthy nutrition and foundational lifestyle behaviours. However, while behavioural outcomes were frequently reported, improvements in BMI were generally modest, suggesting that longer-term, multi-level strategies [52], including food security in regional and remote communities [53], may be necessary to produce sustained physiological changes.

While our findings align with international reviews [54,55,56], this study extends the evidence base by offering a more granular analysis of cultural tailoring mechanisms. Successful interventions often incorporated elements such as Indigenous-led facilitation, integration of traditional knowledge and practices, use of culturally relevant communication styles (e.g., storytelling, visual arts), and community co-design. These features enhanced trust, engagement, and program relevance. In contrast, programs lacking these elements, despite being labelled as culturally appropriate, often struggled with uptake and sustainability [57,58]. Our review also highlights the importance of embedding interventions within broader systems of support, such as schools, families, and Aboriginal health services, which was less emphasised in earlier international reviews.

Despite encouraging outcomes, some interventions reported limited or mixed effects on obesity-related measures. Contributing factors included inadequate community involvement, low fidelity in implementation, and failure to address broader social determinants of health. This pattern has been observed in findings from obesity programs conducted in non-Australian Indigenous children populations such as the Maori New Zealand children programs [59]. Structural barriers, such as food insecurity, limited access to culturally safe healthcare, and unstable housing, remain under-addressed in many programs. These upstream determinants are essential components of a holistic and equitable obesity prevention approach and must be prioritised in future intervention designs. The findings of this review align with international literature on Indigenous child health promotion [60,61]. Consistent with our findings, a systematic review by Littlewood et al. [54], which evaluated six studies targeting Māori and Pacific Islander children in New Zealand, reported limited improvements in anthropometric outcomes. The review went on to highlight the lack of cultural grounding, Indigenous leadership, and co-design principles in these interventions, mirroring gaps observed in several Australian programs.

Similarly, Wahi [55] conducted a review of 13 studies across Canada, Australia, New Zealand, and the United States, focusing on the Indigenous population, and found that short-term, low-intensity programs had minimal impact on anthropometric outcomes, while culturally tailored, long-term community-driven interventions showed greater potential. However, many programs lacked sufficient cultural adaptation or sustainability planning. A Canadian systematic review by Kshatriya [62], which included 10 studies (primarily randomized controlled trials), echoed these concerns. Although peer-led programs that incorporated Indigenous knowledge demonstrated promise, the overall quality of evidence was rated low due to small sample sizes, methodological limitations, and short follow-up periods. The review emphasized the urgent need for rigorous, culturally grounded, community-owned studies to build a stronger evidence base.

Collectively, the evidence underscores the importance of culturally safe, community-driven, and systems-informed approaches to obesity prevention among Indigenous children. Addressing both behavioural and structural determinants of health through co-designed, long-term strategies is essential for achieving sustained improvements in health equity and outcomes.

### 4.1. Strengths and Limitations

The most significant limitation of this review was that it was conducted with a limited number of studies and did not cover all the states of Australia, which could have significantly impacted the outcome.

Using a scoping review created wide coverage of studies highlighting main issues and interventions. This review contributes to an understanding of the importance of health interventions in obesity prevention among ATSI children and adolescents. The review identified an association between PA, healthy diet, and nutrition, as well as behavior change, with decreasing obesity. Additionally, evidence suggests that PA and behavioral change can affect children’s body weight. However, the study was limited due to the lack of literature regarding obesity intervention programs in ATSI communities. Despite a high level of knowledge of nutritious food and PA in children and parents, this knowledge did not generally exist in reports of healthy behavior.

### 4.2. Practice & Policy Implications

Scale and Sustainability: Future interventions should be adequately powered, implemented over longer durations, and geographically inclusive to capture long-term and generalisable impacts.Community-Led Approaches: Programs must be culturally tailored, co-designed with Aboriginal and Torres Strait Islander communities, and embedded in local settings to ensure ownership and effectiveness.Coordination Across Programs: Stronger planning and documentation are needed to avoid overlap with concurrent health initiatives and to clearly identify the effects of individual interventions.Addressing Structural Determinants: Future interventions must go beyond individual behaviour change and explicitly target upstream determinants such as food insecurity, housing instability, and access to culturally safe healthcare. Programs should integrate intersectoral partnerships, linking health, education, housing, and social services to create supportive environments. For example, embedding nutrition programs within school curricula, co-locating health services in community hubs, and ensuring Indigenous leadership in program governance can enhance sustainability and relevance. Funding models should also prioritise long-term investment in community infrastructure and capacity building.

## 5. Conclusions

Childhood obesity prevention interventions in Australia show promising outcomes for Aboriginal and Torres Strait Islander (ATSI) children and adolescents. This scoping review, which analysed eleven studies, found that the most effective programs were community-based; involved multiple stakeholders, including families, educators, and Aboriginal health workers; and incorporated culturally relevant and traditional elements. These findings align with international evidence on Indigenous health, reinforcing the importance of culturally tailored, community-led approaches.

Nevertheless, the small number of eligible studies highlights the need for further research to strengthen the evidence base and confirm associations between intervention characteristics and outcomes. Expanding the scope of future reviews to include additional study designs beyond RCTs and quasi-experimental studies may also provide a more comprehensive understanding and help address existing knowledge gaps.

## Figures and Tables

**Figure 1 ijerph-22-01671-f001:**
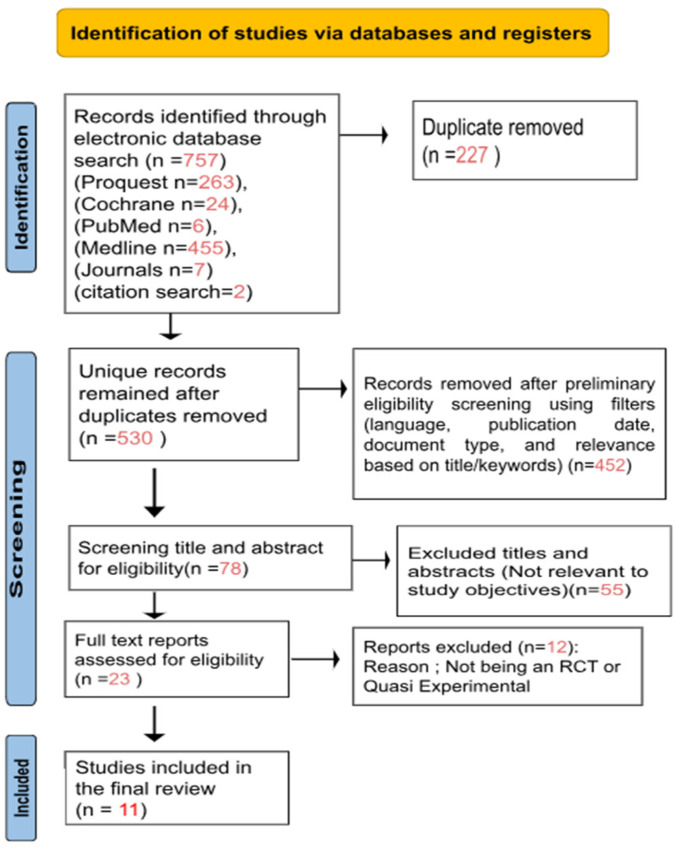
PRISMA Flow Diagram of study selection.

**Table 1 ijerph-22-01671-t001:** Quality appraisal of included studies using CASP criteria.

CASP Checklist Criteria	Allender et al. (2024) [39]	Smithers et al. (2017) [40]	Waters et al. (2018) [41]	Okely et al. (2020) [42]	Bell et al. (2017) [43]	Malseed et al. (2014) [44]	Mihrshahi et al. (2017) [45]	Peralta et al. (2014) [46]	Black et al. (2013) [47]	Pettman et al. (2014) [48]	Browne et al. (2022) [15]
1. Did the study address a clearly focused issue?	✔	✔	✔	✔	✔	✔	✔	✔	✔	✔	✔
2. Was the assignment of participants to interventions randomized?	✔	✔	✔	✔	✘	✘	✘	✘	✘	✘	✘
3. Were all participants who entered the study accounted for at its conclusion?	✔	✔	✔	✔	✔	✔	✔	✔	✔	✔	✔
4. Were participants, staff, and study personnel ‘blind’ to treatment?	✘	✔	✘	✘	✘	✘	✘	✘	✘	✘	✘
5. Were the groups similar at the start of the trial?	✔	✔	✔	✔	X	✔	✔	✔	✔	✔	✔
6. Aside from the intervention, were the groups treated equally?	✔	✔	✔	✔	✔	✔	✔	✔	✔	✔	✔
7. Were all outcomes measured in a reliable way?	✔	✔	✔	✔	✔	✔	✔	✔	✔	✔	✔
8. Was the follow-up of subjects complete and long enough?	✔	✔	✔	✔	✔	✔	✔	✔	✔	✔	✘
9. Were participants analyzed in the groups to which they were randomized?	✔	✔	✔	✔	✔	✔	✔	✔	✔	✔	✘
10. Were results presented with precision (e.g., CI, p-values)?	✔	✔	✔	✔	✔	✔	✔	✔	✔	✔	✔
11. Do the benefits outweigh the harms and costs?	✔	✔	✔	✔	✔	✔	✔	✔	✔	✔	✔
Overall Risk of Bias	Moderate	Low	Moderate	Moderate	Moderate	Moderate	Moderate	High	Moderate	Moderate	Moderate

**Table 2 ijerph-22-01671-t002:** Characteristics and Metadata of Included Studies.

Author, Year, State	Study Title [Name of Trial]	Study Design	Sample Size/Age	Intervention Type	Cultural Tailoring	Community Involvement	Outcome Measures	Summary of Findings	Risk of Bias
[39] Allender et al. 2024 Victoria	Three-year behavioural, health-related quality of life, and body mass index outcomes from the RESPOND randomized trial [RESPOND]	Cluster RCT	5–12 yrs, ATSI & non-ATSI	Community-led systems-based	Yes	Yes	BMI, HRQoL, screen time, water intake, physical activity	Modest improvements in BMI and health behaviours	Moderate
[15] Browne et al. 2022 Victoria	Healthy weight, health behaviours and quality of life among Aboriginal children living in regional Victoria [WHO STOPS & RESPOND]	Cross-sectional analysis	8–13 yrs, ATSI (n = 303)	WHO STOPS & RESPOND data	Yes	Yes	HRQoL, health behaviours	Highlighted disparities; need for culturally appropriate strategies	Moderate
[40] Smithers et al. 2017 South Australia	Diet anAnthropometry at 2 years of age following an oral health promotion programme for Australian Aboriginal children and their carers: [Baby Teeth Talk ]	RCT	6 weeks old (n = 454)	Oral health + dietary advice	Yes	Yes	Diet, anthropometry, health behaviour	Minor dietary improvements, no BMI impact	Low
[42] Okely et al., 2020 NSW	“jump Childcare-based intervention to promote physical activity in preschoolers: Six-month findings from a cluster randomised trial. [Jump Start Trial]	Cluster RCT	3 yrs (n = 658)	Physical activity in childcare	No	Limited	Physical activity	No significant outcomes; fidelity issues	Moderate
[41] Waters et al., 2018. Victoria	CluCl Cluster randomized trial of a school-community child health promotion and obesity prevention intervention: Findings from the evaluation of fun ‘n healthy in Moreland	Cluster RCT	4–13 yrs (n = 2965)	School-community health promotion	No	Yes	Healthy behaviours, anthropometry	Improved behaviours, no BMI change	Moderate
[45] Mihrshahi et al. 2017 Queensland	EvaluI Intervention of the Good Start Program: A healthy eating and physical activity intervention for Maori and Pacific Islander children living in Queensland, Australia	Quasi-experimental	6–19 yrs (n = 375)	Good Start program (performing arts)	Yes	Yes	Nutrition knowledge, physical activity	Improved knowledge and behaviours	Moderate
[43] Bell et al. 2017 South Australia	Changes in weight status, quality of life and behaviours of South Australian primary school children: results from the Obesity Prevention and Lifestyle (OPAL)community intervention program	Quasi-experimental	9–11 yrs (n = 4637)	OPAL systems-wide program	No	Limited	BMI, HRQoL, behaviours	No significant changes	Moderate
[46] Peralta et al. 2014 NSW	Effects of a Community and School Sport-Based Program on Urban Indigenous Adolescents’ Life Skills and Physical Activity Levels: The SCP Case Study [SCP—Sporting Chance Program]	Quasi-experimental	Grades 7–10 (n = 34)	School-community sport program	Yes	Yes	Physical activity, life skills	High acceptability, improved physical activity	High
[44] Malseed et al. 2014 QLD	School Based Health Education Program for Urban Indigenous Young People in Australia [Deadly Choices School Program]	Quasi-experimental	11–18 yrs (n = 103)	Deadly Choices school program	Yes	Yes	Diet, HRQoL, health behaviours	Improved literacy and behaviours	Moderate
[47] Black et al. 2013 NSW	Health Outcomes of a subsidised fruit and vegetable program for Aboriginal children in northern New South Wales [Fruit & Veg Subsidy Trial]	Quasi-experimental	<17 yrs (n = 167)	Fruit & Veg subsidy	Yes	Yes	Diet, anthropometry, haemoglobin	Improved nutrition markers, no BMI change	Moderate
[48] Pettman et al. 2014 South Australia	Findings from the eat well be active community programs [Eat well be active]	Quasi-experimental	0–18 yrs (n = 1062)	Eat Well Be Active (multi-strategy)	Yes	Yes	BMI	Moderate BMI reductions among younger children	Moderate

## Data Availability

All data are available and accessible via the references list of this review.

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
