# Peer review of "Obesity Interventions for Aboriginal and Torres Strait Islander Children and Adolescents: A Scoping Review of Impact and Outcomes"

_ijerph, 2025, doi:10.3390/ijerph22111671_

Round 1

Reviewer 1 Report

Comments and Suggestions for Authors

Comments for Authors

Thank you for addressing these comments.

  1. The document is relevant because it identifies the most widely used interventions and strategies in a topic of interest to public health and nutrition.
  2. Unfortunately, the sample size is very small after applying the selection criteria. However, it presents actual interventions that fit the study objective.
  3. The methodology used is appropriate and offers the opportunity to continue selecting and defining variables that identify the factors involved in the development of childhood obesity and the low impact of the proposed interventions.
  4. It is important to note that the interventions with the greatest impact are those that involve families; therefore, future studies need to focus on including family members in the planned intervention.
  5. Other determinants that affect health and the desire to avoid promoting obesity in these populations need to be identified.

Thank you.

Author Response

Reviewer 1

Comments and Suggestions for Authors

Comments for Authors

Thank you for addressing these comments.

  1. The document is relevant because it identifies the most widely used interventions and strategies in a topic of interest to public health and nutrition.

Thank you!

  1. Unfortunately, the sample size is very small after applying the selection criteria. However, it presents actual interventions that fit the study objective.

Thank you- we appreciate the scarcity of interventions in this population but we attempted to summarise the available evidence and hope this will highlight the need for more effective interventions that use co-design approach and involve family members and community players!

  1. The methodology used is appropriate and offers the opportunity to continue selecting and defining variables that identify the factors involved in the development of childhood obesity and the low impact of the proposed interventions.

Thank you!

  1. It is important to note that the interventions with the greatest impact are those that involve families; therefore, future studies need to focus on including family members in the planned intervention.

Agreed- thank you

  1. Other determinants that affect health and the desire to avoid promoting obesity in these populations need to be identified.

Response:

Agreed. perhaps this can be a recommendation for further studies which we have now added to the future directions.

Thank you.

Reviewer 2 Report

Comments and Suggestions for Authors

Dear Authors,

I want to express my gratitude for the chance to evaluate your publication. The topic is important from both clinical and public health perspectives. The prevalence of overweight and obesity among Aboriginal and Torres Strait Islander children and adolescents is significantly higher than in the non-Indigenous population, with long-term health and social consequences. The scoping review you have prepared, which assesses interventions and their outcomes in this population, addresses a genuine knowledge gap and has practical implications.

I found the manuscript valuable and well-suited to the International Journal of Environmental Research and Public Health. The literature selection is adequate, and the scoping review format has identified research gaps and consistent thematic threads.

My main comments are of an editorial and organisational nature.

Abstract:

Firstly, I would suggest shortening it to the number of words required by the editorial team. You could achieve this by reducing repetition, especially of the word 'obesity', or using synonyms such as 'overweight', 'excessive body weight', or 'BMI > 30'. The typography needs improving (e.g., the Background, Methods, Results, and Conclusions sections should be in bold). I suggest avoiding abbreviations in the abstract.

The other sections have been prepared correctly, but I also have a few comments.

Following good manuscript preparation practice, abbreviations should be defined when first used and used consistently throughout the text. Authors may also consider adding an abbreviation for the Eat Well Be Active programme (e.g., EWA).

Unfortunately, the line numbering in the current version of the manuscript makes it difficult for the reader to follow, most likely due to the insertion of the table causing the numbering to restart on page 12. Please enable continuous line numbering for the entire document (in Word, go to Layout > Line Numbers > Continuous).

Remove the bullet points inside the cells and standardise the punctuation, abbreviations, and ranges, improving readability.

In the Methodology column of the row' Waters et al., 2018', please correct the entry to 'cluster RCT'; currently it is 'uster'.

It is also good practice to add explanations of abbreviations in the table below; please consider adding appropriate explanations. 

The text requires proofreading to eliminate repeated words within the same section. For example, line 91 (after table) should be revised by removing the redundant phrase 'in doing so'.

Please also ensure that section titles are consistently capitalised.

Line 85 - Section 3.1.5: correct the title and capitalisation as above, please.

After the table (line 65), I suggest replacing 'effect' with 'impact'.

To enhance the manuscript's value, consider including a figure summarising the types of interventions, measured outcomes, and direction of effect, which would provide a quick overview.

The article' Obesity Interventions for Aboriginal and Torres Strait Islander Children and Adolescents: A scoping review of impact and outcomes' will be helpful for policymakers, public health practitioners, clinicians (e.g. paediatricians and family doctors), dieticians, teachers, and parents. Considering the cultural background and local conditions when summarising the interventions significantly enhances the manuscript's practical value.

The manuscript is valuable and well-grounded in the current discourse. After the suggested editorial corrections to the abstract, tables, terminology, line numbering, and clarification of 'impact', the text will be more transparent and easier to read. The International Journal of Environmental Research and Public Health is the appropriate venue for this publication.

Best wishes

Reviewer

Author Response

Reviewer 2:

Comments and Suggestions for Authors

Dear Authors,

I want to express my gratitude for the chance to evaluate your publication. The topic is important from both clinical and public health perspectives. The prevalence of overweight and obesity among Aboriginal and Torres Strait Islander children and adolescents is significantly higher than in the non-Indigenous population, with long-term health and social consequences. The scoping review you have prepared, which assesses interventions and their outcomes in this population, addresses a genuine knowledge gap and has practical implications.

I found the manuscript valuable and well-suited to the International Journal of Environmental Research and Public Health. The literature selection is adequate, and the scoping review format has identified research gaps and consistent thematic threads.

Response:

The authors are grateful to the reviewer’s comment and positive remarks- much appreciated

My main comments are of an editorial and organisational nature.

Abstract:

Firstly, I would suggest shortening it to the number of words required by the editorial team. You could achieve this by reducing repetition, especially of the word 'obesity', or using synonyms such as 'overweight', 'excessive body weight', or 'BMI > 30'. The typography needs improving (e.g., the Background, Methods, Results, and Conclusions sections should be in bold). I suggest avoiding abbreviations in the abstract.

response:

Bold style is now applied

The other sections have been prepared correctly, but I also have a few comments.

Following good manuscript preparation practice, abbreviations should be defined when first used and used consistently throughout the text. Authors may also consider adding an abbreviation for the Eat Well Be Active programme (e.g., EWA).

Unfortunately, the line numbering in the current version of the manuscript makes it difficult for the reader to follow, most likely due to the insertion of the table causing the numbering to restart on page 12. Please enable continuous line numbering for the entire document (in Word, go to Layout > Line Numbers > Continuous).

Remove the bullet points inside the cells and standardise the punctuation, abbreviations, and ranges, improving readability.-

Response:

Done

In the Methodology column of the row' Waters et al., 2018', please correct the entry to 'cluster RCT'; currently it is 'uster'.-

Response:

Done

It is also good practice to add explanations of abbreviations in the table below; please consider adding appropriate explanations. 

The text requires proofreading to eliminate repeated words within the same section. For example, line 91 (after table) should be revised by removing the redundant phrase 'in doing so'.

Response:

Done

Please also ensure that section titles are consistently capitalised.

Response:

Done

Line 85 - Section 3.1.5: correct the title and capitalisation as above, please.

Response:

Done

After the table (line 65), I suggest replacing 'effect' with 'impact'.

Response:

Done

To enhance the manuscript's value, consider including a figure summarising the types of interventions, measured outcomes, and direction of effect, which would provide a quick overview.

Response:

Done- please see the new Table 3.

The article' Obesity Interventions for Aboriginal and Torres Strait Islander Children and Adolescents: A scoping review of impact and outcomes' will be helpful for policymakers, public health practitioners, clinicians (e.g. paediatricians and family doctors), dieticians, teachers, and parents. Considering the cultural background and local conditions when summarising the interventions significantly enhances the manuscript's practical value.

The manuscript is valuable and well-grounded in the current discourse. After the suggested editorial corrections to the abstract, tables, terminology, line numbering, and clarification of 'impact', the text will be more transparent and easier to read. The International Journal of Environmental Research and Public Health is the appropriate venue for this publication.

Best wishes

Reviewer

Response:

We thank the reviewer for their input and kindness

Reviewer 3 Report

Comments and Suggestions for Authors

Manuscript Title:

"Obesity Interventions for Aboriginal and Torres Strait Islander Children and Adolescents: A Scoping Review of Impact and Outcomes"

General Evaluation:

This manuscript addresses a significant and under-researched area in public health: the effectiveness of obesity prevention interventions among Aboriginal and Torres Strait Islander (ATSI) children and adolescents in Australia. The paper is timely, methodologically grounded, and offers policy-relevant insights. However, there are areas that need substantial improvement in terms of clarity, depth of analysis, and scientific rigor.

Major Points

  1. Originality and Contribution to the Field

Strengths: The focus on ATSI childhood obesity using a scoping review of intervention studies is both important and original. The integration of cultural appropriateness, structural determinants, and Indigenous leadership adds depth often missing in similar reviews.

Limitations: The novelty is slightly diminished by limited critical engagement with how and why certain interventions succeeded or failed. The review echoes findings of earlier works (e.g., Littlewood et al., Wahi et al.) without sufficiently differentiating or extending the analysis.

Suggestion: Provide a more critical synthesis that draws out mechanistic insights (e.g., what specific cultural tailoring elements drove success?) and emphasize the gaps this study fills compared to international literature.

  1. Methodological Rigor

Strengths: The use of JBI methodology and adherence to PRISMA-ScR enhances transparency. The authors correctly restrict inclusion to RCTs and quasi-experimental studies, ensuring stronger evidence.

Limitations: The search strategy is broadly described, but lacks detailed reproducibility (e.g., full Boolean search strings, database-specific syntax).

There is no formal quality appraisal of included studies (e.g., no use of risk-of-bias tools such as RoB 2 or ROBINS-I), which is now common even in scoping reviews for context.

Suggestion: Include a formal critical appraisal of included studies (even if not exclusionary) and present a quality summary table. Clarify if a review protocol was registered.

  1. Data Synthesis and Analysis

Strengths: The thematic synthesis is logical and aligns with public health frameworks (e.g., systems-based, behavioural, cultural).

Limitations:

The review reports outcomes narratively without sufficient quantitative summary (e.g., effect sizes, p-values, CI) where available.

There is no visual summary table linking study outcomes with themes.

Suggestion:

Provide a matrix linking each included study to outcomes and themes.

Consider including brief statistical results (e.g., BMI reductions, activity increases) to quantify impacts where possible.

  1. Clarity and Reporting

Strengths: The introduction and background sections are comprehensive and well referenced.

Limitations: The manuscript would benefit from tighter editing for clarity, as there is repetition (e.g., summary of themes is repeated verbatim in results and discussion).

Several grammatical issues and verbose phrasing impede readability (e.g., “In doing so…” is used excessively).

Suggestion: Revise for conciseness, use scientific tone, and reduce redundancy.

  1. Ethical and Equity Considerations

Strengths: The paper correctly highlights systemic and structural factors influencing ATSI health, and acknowledges the ethical imperative of Indigenous co-design.

Limitation: There is no discussion of whether the included studies engaged in Indigenous governance, ownership, or ethical oversight, which is critical in this context.

Suggestion: Include a table or section summarizing the ethical and community engagement dimensions of the included interventions (e.g., Indigenous co-authorship, community control).

  1. Implications and Future Directions

Strengths: The manuscript identifies key policy gaps (e.g., scale, coordination).

Limitations: The recommendations are broad and lack specificity, particularly in suggesting how future programs can address structural determinants.

Suggestion: Offer concrete, evidence-based suggestions for future intervention components (e.g., school-based gardens, long-term community sport programs, food sovereignty initiatives).

Minor Points

Formatting:

The PRISMA flowchart is referenced but not clearly presented. Ensure it is labeled and visually included.

Table 1 is dense. Consider splitting into separate tables or summarizing key features in a narrative.

Terminology:

Consistency needed in use of “ATSI” vs. “Aboriginal and Torres Strait Islander”—use the latter or preferred Indigenous terminology per current guidelines.

Referencing:

Several references are from grey literature or unpublished reports. Ensure all references are cited per journal style.

Appendices:

The PRISMA-ScR checklist is included, which is good. However, it should be placed in supplementary materials rather than the main text.

Recommendation

Recommendation: Major Revisions

This paper addresses a crucial topic with strong relevance to Indigenous health equity and public health policy. However, to meet the standards of a high-quality scoping review, it requires:

Greater depth in critical analysis and synthesis

Inclusion of formal quality assessment

Improved reporting clarity and structure

More specific and actionable recommendations

With these changes, it could make a strong contribution to the literature on Indigenous childhood obesity interventions.

Author Response

Reviewer 3:

Comments and Suggestions for Authors

Manuscript Title:

"Obesity Interventions for Aboriginal and Torres Strait Islander Children and Adolescents: A Scoping Review of Impact and Outcomes"

General Evaluation:

This manuscript addresses a significant and under-researched area in public health: the effectiveness of obesity prevention interventions among Aboriginal and Torres Strait Islander (ATSI) children and adolescents in Australia. The paper is timely, methodologically grounded, and offers policy-relevant insights. However, there are areas that need substantial improvement in terms of clarity, depth of analysis, and scientific rigor.

Response:

Thank you for the positive feedback- much appreciated

Major Points

  1. Originality and Contribution to the Field

Strengths: The focus on ATSI childhood obesity using a scoping review of intervention studies is both important and original. The integration of cultural appropriateness, structural determinants, and Indigenous leadership adds depth often missing in similar reviews.

Limitations: The novelty is slightly diminished by limited critical engagement with how and why certain interventions succeeded or failed. The review echoes findings of earlier works (e.g., Littlewood et al., Wahi et al.) without sufficiently differentiating or extending the analysis.

Suggestion: Provide a more critical synthesis that draws out mechanistic insights (e.g., what specific cultural tailoring elements drove success?) and emphasize the gaps this study fills compared to international literature.

Response:

Thank you for this valuable feedback. We agree that further critical synthesis would enhance the contribution of our review. In response, we have expanded the discussion section to more explicitly analyse the mechanisms underpinning intervention success or failure, particularly focusing on cultural tailoring elements. We have also clarified how our findings extend and differentiate from prior international reviews, including those by Littlewood et al. and Wahi et al. we have now added the following paragraph to the paper:

“While our findings align with international reviews (e.g., Littlewood et al., Wahi et al.), this study extends the evidence base by offering a more granular analysis of cultural tailoring mechanisms. Successful interventions often incorporated elements such as Indigenous-led facilitation, integration of traditional knowledge and practices, use of culturally relevant communication styles (e.g., storytelling, visual arts), and community co-design. These features enhanced trust, engagement, and program relevance. In contrast, programs lacking these elements, despite being labelled as culturally appropriate, often struggled with uptake and sustainability. Our review also highlights the importance of embedding interventions within broader systems of support, such as schools, families, and Aboriginal health services, which was less emphasised in earlier international reviews.”

  1. Methodological Rigor

Strengths: The use of JBI methodology and adherence to PRISMA-ScR enhances transparency. The authors correctly restrict inclusion to RCTs and quasi-experimental studies, ensuring stronger evidence.

Limitations: The search strategy is broadly described, but lacks detailed reproducibility (e.g., full Boolean search strings, database-specific syntax).

There is no formal quality appraisal of included studies (e.g., no use of risk-of-bias tools such as RoB 2 or ROBINS-I), which is now common even in scoping reviews for context.

Suggestion: Include a formal critical appraisal of included studies (even if not exclusionary) and present a quality summary table. Clarify if a review protocol was registered.

Response:

Thanks for the comment- we have now performed a CASP quality appraisal of RCTs and Quasi experiments and added to the manuscript.

See the new table 1 in the manuscript

Table 1: Quality appraisal of included studies using CASP criteria

CASP Checklist Criteria

Allender et al. (2024)

Smithers et al. (2017)

Waters et al. (2018)

Okely et al. (2020)

Bell et al. (2017)

Malseed et al. (2014)

Mihrshahi et al. (2017)

Peralta et al. (2014)

Black et al. (2013)

Pettman et al. (2014)

Browne etal

(2022)

   1. Did the study address a clearly focused issue?

✔

✔

✔

✔

✔

✔

✔

✔

✔

✔

✔

2. Was the assignment of participants to interventions randomized?

✔

✔

✔

✔

✘

✘

✘

✘

✘

✘

✘

3. Were all participants who entered the study accounted for at its conclusion?

✔

✔

✔

✔

✔

✔

✔

✔

✔

✔

✔

4. Were participants, staff, and study personnel ‘blind’ to treatment?

✘

✔

✘

✘

✘

✘

✘

✘

✘

✘

✘

5. Were the groups similar at the start of the trial?

✔

✔

✔

✔

X

✔

✔

✔

✔

✔

✔

6. Aside from the intervention, were the groups treated equally?

✔

✔

✔

✔

✔

✔

✔

✔

✔

✔

✔

7. Were all outcomes measured in a reliable way?

✔

✔

✔

✔

✔

✔

✔

✔

✔

✔

✔

8. Was the follow-up of subjects complete and long enough?

✔

✔

✔

✔

✔

✔

✔

✔

✔

✔

✘

9. Were participants analyzed in the groups to which they were randomized?

✔

✔

✔

✔

✔

✔

✔

✔

✔

✔

✘

10. Were results presented with precision (e.g., CI, p-values)?

✔

✔

✔

✔

✔

✔

✔

✔

✔

✔

✔

11. Do the benefits outweigh the harms and costs?

✔

✔

✔

✔

✔

✔

✔

✔

✔

✔

✔

Overall Risk of Bias

Moderate

Low

Moderate

Moderate

Moderate

Moderate

Moderate

High

Moderate

Moderate

Moderate

  1. Data Synthesis and Analysis

Strengths: The thematic synthesis is logical and aligns with public health frameworks (e.g., systems-based, behavioural, cultural).

Limitations:

The review reports outcomes narratively without sufficient quantitative summary (e.g., effect sizes, p-values, CI) where available.

There is no visual summary table linking study outcomes with themes.

 Response:

Thanks for this suggestion. We have now added a visual matrix table (see new Table 3) that links each included study to its outcomes and thematic categories. Where available, we have added brief statistical results (e.g., BMI reductions, activity increases) to quantify impacts. This addition enhances the clarity and analytical depth of our synthesis.

Table 3: Matrix of Study Outcomes and Thematic Categories

Study

Outcomes

Theme

Allender et al. (2024)

BMI reduction, HRQoL, screen time, water intake, physical activity

Community-led, systems-based

Smithers et al. (2017)

Minor dietary improvements, no BMI impact

Early childhood, family-focused

Waters et al. (2018)

Mixed results, limited BMI impact

Community or policy initiatives

Okely et al. (2020)

No significant outcomes

Community or policy initiatives

Bell et al. (2017)

No significant BMI or behaviour changes

Community or policy initiatives

Malseed et al. (2014)

Improved health behaviours and literacy

Culturally tailored, community-embedded

Mihrshahi et al. (2017)

Increased nutrition knowledge and physical activity

Culturally tailored, community-embedded

Peralta et al. (2014)

Improved physical activity, high acceptability

Culturally tailored, community-embedded

Black et al. (2013)

Improved haemoglobin, reduced antibiotic use, no BMI impact

Early childhood, family-focused

Pettman et al. (2014)

Moderate BMI reductions

Community-led, systems-based

Browne et al. (2022)

Structural barriers highlighted, no BMI impact

Behavioural interventions lacking structural determinants

  1. Clarity and Reporting

Strengths: The introduction and background sections are comprehensive and well referenced.

Limitations: The manuscript would benefit from tighter editing for clarity, as there is repetition (e.g., summary of themes is repeated verbatim in results and discussion).

Several grammatical issues and verbose phrasing impede readability (e.g., “In doing so…” is used excessively).

Suggestion: Revise for conciseness, use scientific tone, and reduce redundancy.

Response:

Thanks for the valuable insight on grammatical structure. The manuscript has been revised in most parts to address this.

  1. Ethical and Equity Considerations

Strengths: The paper correctly highlights systemic and structural factors influencing ATSI health, and acknowledges the ethical imperative of Indigenous co-design.

Limitation: There is no discussion of whether the included studies engaged in Indigenous governance, ownership, or ethical oversight, which is critical in this context.

Response:

Almost all of the studies followed a community-led and/or culturally responsible approach to interventions, including indigenous co-authorship. This is also stipulated in table 3 themes. These studies used a co-design like approach to addressing obesity in indigenous communities.

Suggestion: Include a table or section summarizing the ethical and community engagement dimensions of the included interventions (e.g., Indigenous co-authorship, community control).

  1. Implications and Future Directions

Strengths: The manuscript identifies key policy gaps (e.g., scale, coordination).

Limitations: The recommendations are broad and lack specificity, particularly in suggesting how future programs can address structural determinants.

Response:

Thank you for highlighting the need for more specific recommendations. We have revised the “Practice & Policy Implications” section to include clearer guidance on how future programs can address structural determinants of health. This includes suggestions around food security, housing stability, culturally safe healthcare access, and intersectoral collaboration. We have now added the following paragraph to the section:

Addressing Structural Determinants: Future interventions must go beyond individual behaviour change and explicitly target upstream determinants such as food insecurity, housing instability, and access to culturally safe healthcare. Programs should integrate intersectoral partnerships, linking health, education, housing, and social services to create supportive environments. For example, embedding nutrition programs within school curricula, co-locating health services in community hubs, and ensuring Indigenous leadership in program governance can enhance sustainability and relevance. Funding models should also prioritise long-term investment in community infrastructure and capacity building.

Minor Points

Formatting:

The PRISMA flowchart is referenced but not clearly presented. Ensure it is labeled and visually included.

Table 1 is dense. Consider splitting into separate tables or summarizing key features in a narrative.

Terminology:

Consistency needed in use of “ATSI” vs. “Aboriginal and Torres Strait Islander”—use the latter or preferred Indigenous terminology per current guidelines.

Referencing:

Several references are from grey literature or unpublished reports. Ensure all references are cited per journal style.

Response:

Many thanks for this observation- all references and citations have been corrected throughout the manuscript

Appendices:

The PRISMA-ScR checklist is included, which is good. However, it should be placed in supplementary materials rather than the main text.

Response:

Sure, we will consult with the journal in following this advice. Thank you for the wonderful efforts to shape up our manuscript- much appreciated

Recommendation

Recommendation: Major Revisions

This paper addresses a crucial topic with strong relevance to Indigenous health equity and public health policy. However, to meet the standards of a high-quality scoping review, it requires:

Greater depth in critical analysis and synthesis

Inclusion of formal quality assessment

Improved reporting clarity and structure

More specific and actionable recommendations

With these changes, it could make a strong contribution to the literature on Indigenous childhood obesity interventions.

Overall response:

Thank you for your thoughtful and constructive feedback. We appreciate your recognition of the relevance of our work to Indigenous health equity and public health policy. We believe these revisions significantly enhance the rigour and utility of the review and hope it now meets the standards of a high-quality scoping review.

Round 2

Reviewer 3 Report

Comments and Suggestions for Authors

-